# Site Index Estimation Using Airborne Laser Scanner Data in *Eucalyptus dunnii* Maide Stands in Uruguay

Iván Rizzo-Martín [1,*], Andrés Hirigoyen-Domínguez [2], Rodrigo Arthus-Bacovich [3], Mª Ángeles Varo-Martínez [4] and Rafael Navarro-Cerrillo [4]

1 Department of Forest Production and Wood Technology, Faculty of Agronomy, University of the Republic, Montevideo 12900, Uruguay
2 National Institute of Agricultural Research (Instituto Nacional de Investigación Agropecuaria—INIA Uruguay), Tacuarembó, Ruta 5 km 386, Tacuarembó 45000, Uruguay; andreshirigoyen@gmail.com
3 Observatory of Global Change of the Mediterranean Forest, Department of Forest Engineering, University of Córdoba, E-14071 Córdoba, Spain
4 Department of Forestry Engineering, Laboratory of Silviculture, Dendrochronology and Climate Change, DendrodatLab-ERSAF, University of Cordoba, Campus de Rabanales, Crta. IV, km. 396, E-14071 Córdoba, Spain
* Correspondence: ivan-rizzo@hotmail.com

**Abstract:** Intensive silviculture demands new inventory tools for better forest management and planning. Airborne laser scanning (ALS) was shown to be one of the best alternatives for high-precision inventories applied to productive plantations. The aim of this study was to generate multiple stand-scale maps of the site index (SI) using ALS data in the intensive silviculture of *Eucalyptus dunnii* Maide plantations in Uruguay. Forty-three plots (314.16 m$^3$) were established in intensive *E. dunnii* plantations in the departments of Río Negro and Paysandú (Uruguay). ALS data were obtained for an area of 1995 ha. Linear and Random Forest models were fitted to estimate the height and site index, and OrpheoToolBox (OTB) software was used for stand segmentation. Linear models for dominant height (DH) estimation had a better fit (R$^2$ = 0.84, RMSE = 0.94 m, MAPE = 0.04, Bias = 0.002) than the Random Forest (R$^2$ = 0.85, RMSE = 1.27 m, MAPE = 7.20, Bias=−0.173) model when including only the 99th percentile metric. The coefficient between RMSE values of the cross-validation and RMSE of the model had a higher value for the linear model (0.93) than the Random Forest (0.75). The SI was estimated by applying the RF model, which included the ALS metrics corresponding to the 99th height percentile and the 80th height bicentile (R$^2$ = 0.65; RMSE = 1.62 m). OTB segmentation made it possible to define a minimum segment size of 2.03 ha (spatial radius = 30, range radius = 1 and minimum region size = 64). This study provides a new tool for better forest management and promotes the need for further progress in the application of ALS data in the intensive silviculture of *Eucalyptus* spp. plantations in Uruguay.

**Keywords:** LiDAR; *Eucaliptus* spp.; site Index; random forest; OrpheoToolBox; stand segmentation; precision silvicultural

## 1. Introduction

Uruguay has a surface area of 176,251 km$^2$, of which 66% is currently grasslands (natural, fertilized, improved, or implanted) and 4.77% is occupied by native forests and 5.91% by planted forests [1]. Commercial forest plantations in Uruguay are formed mainly by species from the *Eucalyptus* spp. and *Pinus* spp. genii [2,3]. The predominant eucalyptus species are *Eucalyptus grandis* Hill ex Maide and *Eucalyptus dunnii* Maide, while the predominant species of pines are *Pinus taeda* L. and *Pinus elliottii* Engelm [1]. *Eucalyptus grandis* is used for cellulose pulp and for sawmilling, while *E. globulus* and *E. dunnii* are used only for pulp. In contrast, the *Pinus* is intended for sawmilling purposes only. Rotation of *Eucalyptus* spp. in Uruguay is between 8 and 10 years for pulp purposes and 20 years for

sawmilling. In the case of *Pinus* spp., the rotation corresponds to approximately 25 years for sawmills [1]. The potential productivity of an area is determined by forest site quality, which refers to the volume of wood growth by a forest during the final harvesting. This potential productivity is quantified using the mean dominant height (DH) (e.g., average height of a fixed number of trees per stand with the largest diameters at breast height), with the site index (SI) being an indication of the productive capacity of the site [4]. There are site index estimation equations for different forest species such as *Eucalyptus* ssp., *Picea* ssp. and *Pinus* ssp. at international level [5,6]. In Uruguay, site index estimation equations are available for certain forest species, such as *E. dunnii*, *E. grandis*, *E. globulus* and *Pinus* spp. [7]. These equations consider the current tree age, reference age and current DH [7].

SI estimation has traditionally been conducted using field inventories based on forest variables such as volume, total biomass, basal area, and density, which have different levels of uncertainty and precision [8]. However, field inventories have important limitations when it is necessary to study large areas [9,10]. Remote sensing technologies, which include the use of spatial sensors, aerial orthophotographs and other intensive data collection methods have been generalized to complement fieldwork [11], making it possible to estimate tree and forest variables with lower economic costs, less time invested and less estimation error [12]. Airborne laser scanning (ALS) technology can represent the three-dimensional structure of forests, thus improving the estimation of variables such as biomass, volume or basal area when compared to other two-dimensional measurement sensors such as photographic systems or radiometers [13–16]. In a study conducted in Uruguay, ALS metrics have been used to improve inventories of forest stands of *Eucalyptus* spp. [17].

Stand delineation is complementary to site index and is crucial for efficient forest planning. Forest stands are uniform in composition, size or age, and are managed as a single unit [18]. Stand delineation has traditionally been conducted manually using field information and high-resolution photographic images [14,19]. This method is not efficient because it is time-consuming in regard to analysis and is limited by the degree of subjectivity of different operators [15]. Automatic segmentation based on tree and forest variables derived from ALS metrics is currently available and is useful for precision forestry. Automatic segmentation can generate more homogeneous stands than those defined manually using traditional methods, at lower cost [17,20,21].

However, there is less experience in the use of ALS data to conduct site quality analysis at the stand scale [22]. The use of height projection equations allows the estimation of site index for the reference age, and ALS can be used in these equations. In previous studies, site index was estimated using dynamic SI equations generated with the dominant mean height model based on the use of ALS metrics as independent variables [9,23,24]. The hypothesis of this work was that the segmentation of *Eucalyptus dunnii* stands according to DH would make it possible to infer the site index and mapping site quality for use in forest management. The aim was to generate a stand-scale mapping of the SI for commercial plantations of *Eucalyptus dunnii* Maide using ALS data in fitted parametric and non-parametric models to improve and optimize forest management in Uruguay. The specific objectives were: (i) to estimate the DH of *E. dunnii* plantations using ALS data; (ii) to assess the site index based on ALS DH and stand age; and (iii) to delineate uniform stands on the basis of SI. The methodology developed in this work provides accurate estimates and mappings of DH and SI at the stand scale based on ALS data using an automatic segmentation method. This is an especially useful silvicultural tool with which forest management in terms of harvesting and future plantations can be improved.

## 2. Materials and Methods

### 2.1. Study Sites

This work was conducted in commercial *Eucalyptus dunnii* plantations belonging to the Forestal Oriental S.A. company and located in the departments of Río Negro and Paysandú (Uruguay, 32°33′04.51″ S–57°14′39.61″ W, Figure 1). The climate in the area is, according to Koppen-Geiger's classification, Cfa, which is characterized by hot summers and rainfall

distributed throughout the year, with a mean annual temperature of 19.2 °C and an annual accumulated precipitation of 1262.5 mm [25]. The dominant soil is phaeozem with a mollic horizon within a secondary calcium carbonate layer, and an accumulation of organic matter that is saturated at its bases in its first meter of depth [26]. According to the classification provided by the Uruguayan National Commission for Agro economic Studies of the Land (CONEAT), the predominant soils in the area are characterized by sandy loam to sandy clay loam texture, average to low fertility, moderate depth and the fact that they are generally well drained. This soil group has an average productivity when compared to the forest priority soil groups [3].

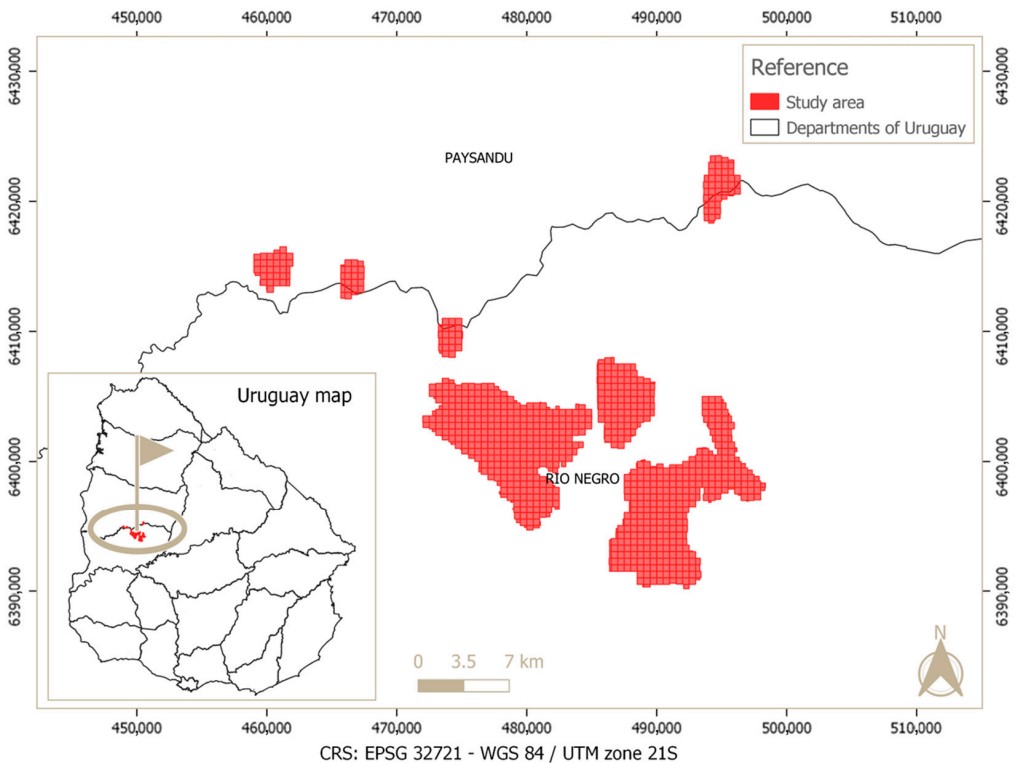

**Figure 1.** Study area of commercial *Eucalyptus dunnii* plantations in Uruguay.

### 2.2. Field Data

In May and June of 2017, 43 plots with a radius of 10 m (314.16 m$^2$) were established in the field. A systematic sampling design was conducted in accordance with traditional inventory procedures for monospecific plantations of *Eucalyptus* sp. Each field plot contained data corresponding to diameter at breast height (dbh at 1.3 m, cm), density (number of trees per hectare), basal area (G, m$^2$ ha$^{-1}$), DH (m, e.g., based on the measurement of ten of the highest trees) and volume per hectare (m$^3$ ha$^{-1}$) (Table 1). The age of trees at each plot was determined through the exact date of planting.

**Table 1.** Silvicultural variables of *Eucalyptus dunnii* plantations in Uruguay. Number of plots (n), age (years), dominant height (DH, m), plot-wise mean diameter at breast height (dbh, cm), basal area (G, m$^2$ ha$^{-1}$), volume (V, m$^3$ ha$^{-1}$) and density (N, trees ha$^{-1}$).

| | Variable | Stdev | Min | Max | Mean |
|---|---|---|---|---|---|
| | Age | 1.71 | 5.00 | 10.00 | 7.44 |
| | DH | 2.43 | 16.20 | 25.30 | 20.33 |
| | dbh | 2.39 | 12.65 | 22.99 | 17.29 |
| *Eucalyptus dunnii* (n = 43) | G | 5.44 | 14.11 | 36.84 | 149.19 |
| | V | 37.87 | 80.49 | 235.94 | 149.19 |
| | N | 231 | 445 | 1401 | 992 |

### 2.3. ALS Data Acquisition and Processing

ALS data were obtained in April 2017 for an area of 1995 ha (Figure 1) using a Riegl VUX-1 laser scanner (RIEGL's Laser measurement systems, Horn, Germany) installed on an autogyro helicopter at a flight altitude of 110 m above ground level, a pulse repetition rate of 550 kHz, a wide angular pitch of 0.0687°, a FOV of 55°, and a point density of 12 points m$^{-2}$. Plots were georeferenced at the WGS84 UTM 21 (EPSG: 32721) coordinate system. The ALS point cloud was processed with the LAStools [27] software using the Windows cmd console to generate the normalized point cloud, digital surface model (DSM) and digital vegetation model (DVM). The pulse distribution was evaluated by employing the *lasgrid* function, which allowed the generation of raster files in which the digital value of each pixel (1 m) corresponds to the density of pulses per m$^2$. The metadata information of the total normalized point cloud was then obtained by implementing the "lasinfo" function, as a result of which the average number of points per m$^2$ was highlighted. Rasters of ALS metrics were obtained using the *lascanopy* function, with a pixel size of 17.7 m, corresponding to the square root of the surface area of the field plot (314.16 m$^2$). The input files used were the normalized point cloud in LAZ format and a vector file in shapefile format that corresponded to the area occupied by the field plots. The parameters "cover_cutoff 2.0" and "height_cutoff 0" were used to consider the points over two meters in height so as to avoid points from non-relevant vegetation, such as scrub, and to obtain percentiles and bicentiles (e.g., 10, 20, 30, 40, 50, 60, 70, 80 and 90) of the entire range of heights.

### 2.4. ALS Estimation Models

First, a correlation matrix between DH and ALS metrics was calculated, and normality of the uncorrelated variables was evaluated by means of the Shapiro–Wilk test [28]. Linear estimation models with ALS metrics taken as the independent variable and DH and SI as the dependent variables were then fitted. Linear models were generated using the *lm* function available in R. The *summary* function in R was used to obtain the intercept, slope, and coefficient of determination (R$^2$). The *RMSE* function of the *MLmetrics* library was utilized to calculate the RMSE of the models. The linear model equation was applied to the selected ALS metrics to obtain a canopy height raster.

The *k*-NN algorithm with Random Forest distance calculation (from here on, *k*-NN-Random Forest) was then applied. In this approach, a Random Forest model is trained with a given dataset, and a proximity matrix is computed based on the frequency that two observations are assigned to the same terminal node (leaf) across all trees in the forest. The proximity matrix provides a representation of the distance or similarity between all pairs of observations in the dataset. The Variance Inflation Factor (VIF) was implemented, and the correlated variables were eliminated using the *varSelection* function of *yaImpute* [29] in the R package with a critical threshold of VIF > 10 [28]. The database was divided into two groups corresponding to training and validation, which contained 70% and 30% of the total data, respectively. The model fit was calculated using the training data, and the *k* value for the Random Forest imputation method was calculated. The *q* value was also calculated to detect a possible overfitting of the model [30]. The *k* value was chosen considering the lowest root mean square error (RMSE). Models were calculated with the *lm* and *yaImput* functions and *randomForest*, *caret* and *yaimute* packages of R software [31–33].

### 2.5. Model Assessment and Validation

Prior to fitting the model k-NN, the most favourable value of k was tested. This was varied in the range between 1 and 11 by using the *caret* library in R. The most favourable *k* was subsequently employed to carry out the imputation of the *k*-NN model with the distance calculated by Random Forest.

The R$^2$ was used to determine the goodness of fit of the models k-NN and the linear models. The predictive performance of the models generated was evaluated by studying the root mean square error (RMSE), the mean absolute percentage error (MAPE) and the model bias (Bias). Simple linear regression related observed and predicted values were

evaluated using the $R^2$ to assess the models selected [34]. After the best model was selected, it was validated using one cross-validation technique [17,35,36]. The $R^2$, RMSE and Bias values were calculated for each fit model and were averaged to obtain the $R^2$, RMSE and Bias from the cross-validation, which were compared with the values obtained for the model selected [34]. The model with the best fit corresponds to that with close values of RMSE of the cross-validation and the RMSE of the evaluated model [34].

When validating the non-parametric *k*-NN Random Forest model, the internal (training) and external (evaluation) accuracy was studied [37] using the calibration and validations databases independently, and the estimation errors were calculated. In this case, the quality of the validation estimates is higher when the ratio between the RMSE of the training data and the RMSE of the evaluation data is closer to 1. The best linear and *k*-NN Random Forest models were compared by employing $R^2$ and RMSE [37].

### 2.6. Canopy Height and Site Index Rasters

To generate the canopy height raster, the best model was applied using the ALS metrics at a pixel size of 17.7 cm, corresponding to the plot surface (314.16 m$^2$). The pixel size of the ALS raster was determined by applying the square root of the area of the field plot to improve accordance between both data. This raster had a value of the average canopy height (m) for each pixel.

A new attribute corresponding to the site index value was generated in the database using the equation proposed for *E. dunnii* [7]. This equation considered the DH of the plot ($DH_1$), the age of the tree ($t_1$ based on verified date of plantation) and the age of the reference rotation ($t_2$). In the case of *E. dunnii*, a $t_2$ equal to 8 years was considered [7]. Linear and non-parametric models were then generated to estimate site index as a function of the ALS metrics. The best of these models was applied to the ALS metrics raster to obtain the site index raster (Equation (1)):

$$DH_2 = DH_1 \times \frac{\left[1 - e^{[-0.15 \times t_2]}\right]^{1.0915}}{\left[1 - e^{[-0.15 \times t_1]}\right]^{1.0915}}, \tag{1}$$

where $DH_1$ is dominant height (m) at time $t_1$ (years) and $DH_2$ is dominant height (m) projected (SI) at time $t_2$, which is the reference age of the species (years).

### 2.7. Segmentation Method

The algorithm used to conduct the stand segmentation was based on the non-parametric estimator Mean Shift (MS) implemented in the OrpheoToolBox (OTB) software available in QGIS tools [38]. This algorithm groups adjacent pixels of a raster that have similar spectral characteristics into segments. This automatic segmentation method was applied to the SI raster to generate homogeneous segments in terms of site index values. In this work, the function *otbcli_LargeScaleMeanShift* of the OTB tool was used. The parameters corresponded to spatial radius (SR), range radius (RR) and minimum region size (MRS). The value of SR was used to define the neighborhood, while the value of RR refers to the digital value threshold that the algorithm considers to be delimiting segments, and the value of MRS defines the smallest size of the segment [39,40]. These parameters define the homogeneity within each segment and the heterogeneity between different segments. In this work, different combinations of parameters were evaluated, which led to the generation of different segmentations. The set of parameters that resulted in the lowest intra-segment variation and the highest inter-segment variation was chosen. This variation was obtained by means of the statistical zonal tool from the QGIS software, taking as input the shapefile resulting from the segmentation.

The segmentation method applied to the SI raster was assessed using an unsupervised evaluation (NSE) technique. NSE generates results of a higher efficiency and a lower subjectivity than the supervised techniques [41–43]. The best segmentation was selected by calculating the internal homogeneity and heterogeneity between segments. The internal

homogeneity of segments was calculated by selecting the lower variance and lower difference between variances. Heterogeneity between segments was determined using the mean variance of site index values of each segment (i.e., segment with the highest variance value was selected as being the best segmentation).

## 3. Results

### 3.1. Linear Model with Which to Estimate Height

Linear models to estimate the DH for *Eucalyptus dunnii* selected 99th, 95th and 90th height percentiles. The models had similar $R^2$ and a slight variation in the RMSE. The linear model with the 99th height percentile (Model 1, Table 2) was, therefore, selected since there was a higher ratio between the RMSE of the model and the RMSE of the cross-validation (RMSE/RMSEcv = 0.93). Figure 2 shows the relationship between the observed and predicted DH values for the model selected (Model 1; $R^2 = 0.82$), considering a significance of *p*-value < 0.01.

**Table 2.** Linear and Random Forest models to estimate dominant height for *Eucalyptus dunnii*. DH (m): dominant height; $R^2$: coefficient of determination; RMSE: root mean square error (m); MAPE: mean absolute percentage error; Bias: model bias; RMSEcv, MAPEcv and BIAScv systematic errors obtained in cross validation.

|  | Models | $R^2$ | RMSE | MAPE | Bias | RMSE cv | MAPE cv | BIAS cv | RMSE/ RMSEcv |
|---|---|---|---|---|---|---|---|---|---|
|  | | | | | Linear models | | | | |
| DH (m) | 5.96 + 0.659 *p99 Model 1 | 0.84 | 0.94 | 0.04 | 0.002 | 1.00 | 0.83 | <0.001 | 0.94 |
|  | 5.442 + 0.721 *p95 Model 2 | 0.85 | 0.90 | 0.04 | −0.02 | 1.16 | 0.97 | 0.01 | 0.77 |
|  | 4.444 + 0.794 *p90 Model 3 | 0.85 | 0.99 | 0.41 | 0.001 | 2.04 | 1.22 | 0.01 | 0.49 |
|  | | | | | Random Forest | | | | |
| DH (m) | Model 4 | 0.85 | 1.27 | 7.20 | −0.173 | 2.12 | 0.60 | | |

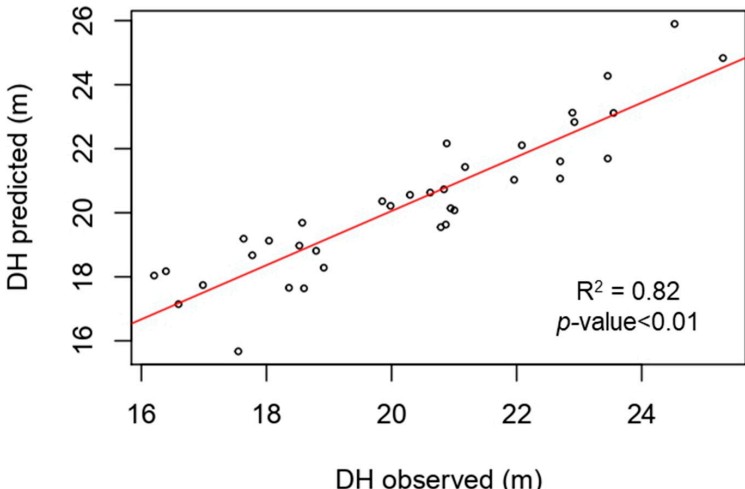

**Figure 2.** Relationship between observed and predicted values for DH (m) of *Eucalyptus dunnii* plantations in Uruguay obtained using Model 1. $R^2$ and *p*-value are included.

### 3.2. k-NN Random Forest Model with Which to Estimate Height

Candidate metrics as predictor variables of DH were defined using the Pearson correlation. The metrics selected were used together and individually within the model, with the 99th percentile model having the highest $R^2$ value ($R^2 = 0.85$, RMSE = 1.27 m, Table 2), using a value of k equal to five. In addition, the 99th percentile was presented a Pearson correlation of 0.92 with DH. The value of q in this model corresponded to a value of 1.26, which had some overfitting. The cross-validation had an RMSEcv value of 1.67, and a ratio

between the RMSE of the model and the RMSE of the cross-validation equal to 0.76. The value of the ratio between internal precision and external precision was 0.60 (Table 2).

Figure 3 shows the relationship between the observed and the predicted DH for Model 4 ($R^2 = 0.57$, *p*-value < 0.01).

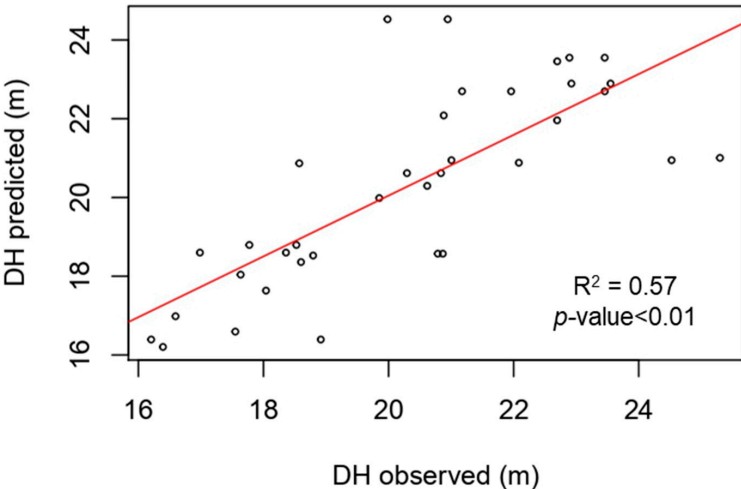

**Figure 3.** Relationship between observed and predicted values for DH (m) of *Eucalyptus dunnii* plantations in Uruguay obtained using Model 4. $R^2$ and *p*-value are included.

### 3.3. Site Index and Height Raster

Two rasters of stand height were obtained, one using Model 1 and the other using Model 4, and they were applied for the estimation of SI. When using Model 1 (linear model), the estimation of site index did not fit satisfactorily. Model 4 (*k*-NN Random Forest) had a $R^2 = 0.65$, an RMSE of 1.62 m and a Bias of −0.3 when using the 99th height percentile and the 80th height bicentile as independent variables. This model used a value of *k* of 3, and the value of *q* of 1.33, which had some overfitting. Figure 4 shows the relationship between the observed and the predicted SI values for the *k*-NN Random Forest model. The raster stack of the 99th percentile raster and the 80th bicentile raster was then used to produce a single two-band raster, maintaining the digital levels of the metrics in each band. The *k*-NN Random Forest model was then applied to this raster stack to obtain the SI raster.

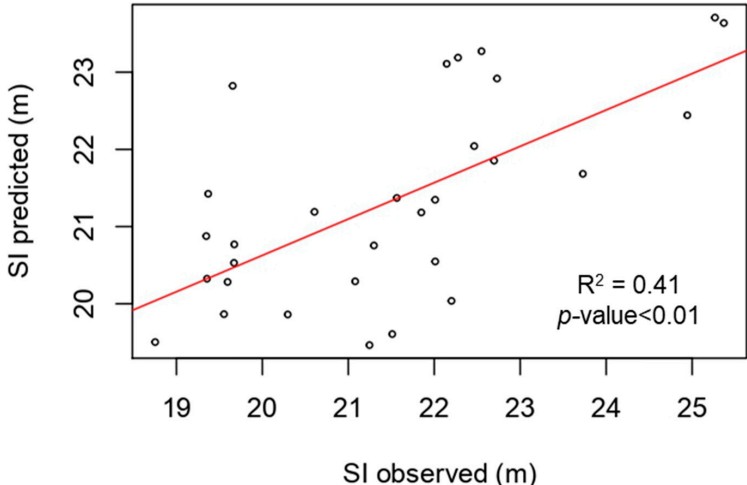

**Figure 4.** Relationship between observed and predicted values for site index (SI m) of Eucalyptus dunnii plantations by Random Forest.

### 3.4. Segmentation OTB

OTB segmentation for forest stand delineation based on site index varied according to the combination of SR, RR and MRS. Various combinations of these parameters were evaluated, and the ten best segmentations were selected (Table 3). The "Hete. ranking" obtained the greatest variance between the site index means of each segment, while the "Homo. ranking" obtained the greatest variance in the site index internal variance values of each segment. The segmentation selected corresponded to that with the lowest intra-segment variation and the highest inter-segment variation. The parameters of this segmentation corresponded to a value of 30 for SR, 1 for RR and 64 for MRS (Table 3). The value of 1 for RR defined the interval in the spectral space when performing the segment delineation, showing important differences in SI among segments. Since a pixel equals 314 m$^2$, the MRS value of 64 was equivalent to a segment area of 2.03 ha. The segmentation presented 585 stands for the total surface (1995 ha) with an average area of 11.6 ha (Table 3).

Figure 5 represents the stand delimitation map based on the OTB segmentation as a function of the SI.

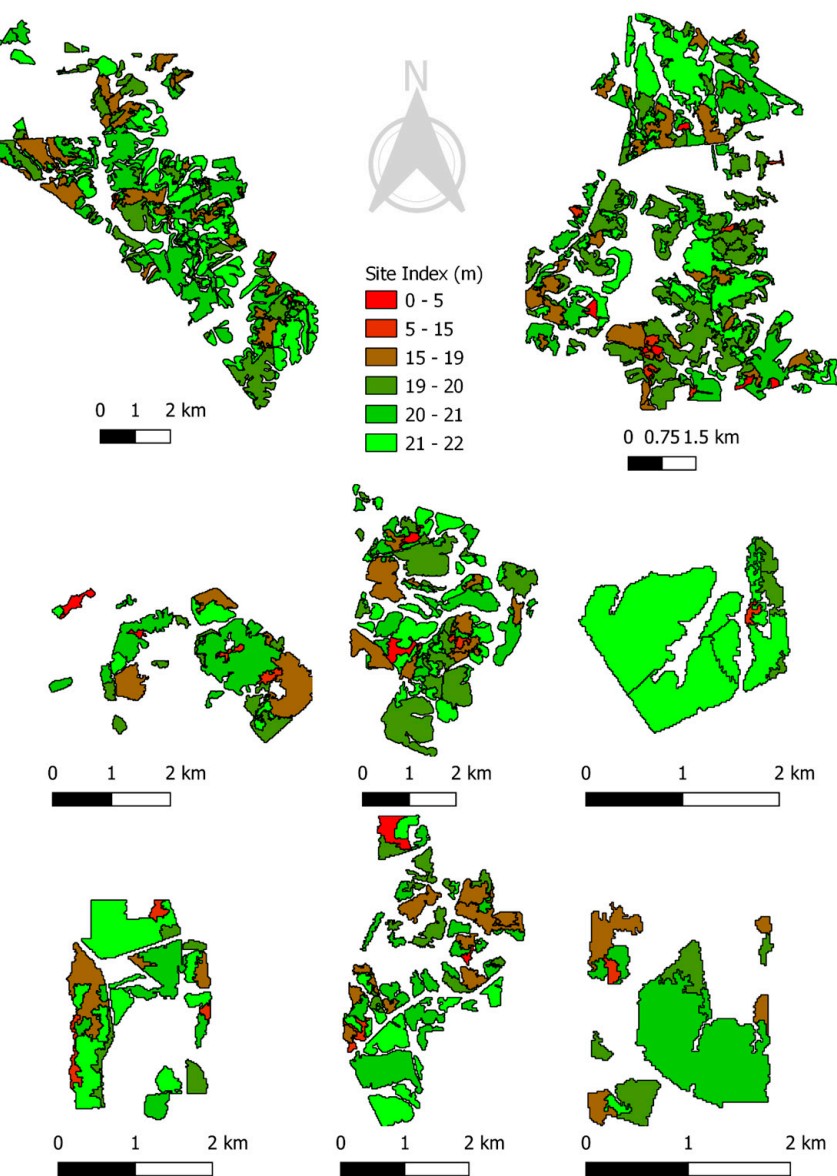

**Figure 5.** Representation of the detail of *Eucalyptus dunnii* plantation segments of segmentation "i" by site index.

**Table 3.** Combination of spatial radius, range radius and minimum region size parameters and evaluation of segmentations: Seg.: segmentation name; SR: spatial radius; RR: range radius; MRS: minimum region size; Nº seg.: number of segments.; Area (ha): mean area of segments in hectares.

| Seg. | Parameters | | | Nº Seg. | Area (ha) |
|------|------|------|------|---------|-----------|
| | SR | RR | MRS | | |
| a | 20 | 8 | 16 | 529 | 10.81 |
| b | 20 | 3 | 16 | 582 | 11.77 |
| c | 16 | 3 | 16 | 635 | 10.96 |
| d | 30 | 1 | 64 | 585 | 11.6 |
| e | 20 | 1 | 16 | 1558 | 4.39 |
| f | 35 | 1 | 64 | 584 | 11.63 |
| g | 4 | 3 | 16 | 413 | 16.63 |
| h | 20 | 1 | 64 | 630 | 10.79 |
| i | 20 | 1 | 95 | 499 | 13.55 |
| j | 35 | 1 | 95 | 458 | 14.74 |

## 4. Discussion

Technology involving ALS sensors is widely applied in forest inventories to determine forest variables [13,44]. In this work, high spatial resolution SI and DH rasters were generated using models based on ALS metrics. The estimation of DH is closely related to the SI, and both are used as indicators of forest site quality [43]. Linear models and the *k*-NN algorithm with Random Forest imputation were compared. Linear models were used because of their simplicity and the fact that less processing is required when generating maps of variables, and the non-parametric model was applied owing to its quality of being independent of data distribution. The Mean Shift segmentation algorithm available in the OTB software was used for stand delineation based on the SI, thus making it possible to delimit areas of different forest productivity for *E. dunnii* plantations.

### 4.1. Height Estimation Models and Generation

Methods most employed to estimate height in forest inventories using ALS metrics correspond to simple regression methods and non-parametric methods [45,46]. Our results showed that both methods were suitable for estimating DH based on ALS metrics. Linear models selected as being the best ALS predictors of DH were the 99th, 95th and 90th percentiles. This is because these percentiles correspond to the first returns of an ALS point cloud generated by the impact of one laser pulse on the highest part of the tree canopy. This is reflected in a high Pearson correlation between DH and these percentiles. Several studies have confirmed a high correlation between dominant tree height and metrics corresponding to the highest percentiles [14,17,41,47]. Our models (Model 1, 2, 3 and 4) had a high accuracy with low RMSE and high $R^2$ values, and a high ratio between the RMSE of the model and the RMSE of the cross-validation (Table 2), as observed in previous work [14,17,41,47]. Random Forest imputation has also been highlighted as being one of the best *k*-NN models to estimate DH, total volume, and biomass. [17,37]. Furthermore, *k*-NN Random Forest model used the 99th percentile as the only independent variable. When comparing linear and Random Forest imputation models, a similar value of $R^2$ and RMSE was observed, but the *k*-NN Random Forest model had a slightly higher RMSE value when compared to the linear models. It should also be noted that the lineal model including the 99th percentile had a higher ratio between the RMSE of the model and the RMSE of the cross-validation (0.94). Moreover, the relationship between the observed and the predicted values had a higher determination coefficient for this model ($R^2 = 0.82$) than for the *k*-NN Random Forest model ($R^2 = 0.57$). These results coincide with precedents regarding the comparison of linear and non-parametric models [17,48]. However, this work highlights the inclusion of only one ALS metric (99th percentile) in the *k*-NN Random Forest model when compared to previous studies in Uruguay which included the 75th percentile and Elev.max metrics (maximum statistic of all heights above the cut-off height of the point cloud) [17].

Height maps generated with lineal and *k*-NN Random Forest models showed a high concordance with ground values represented by the 99th percentile raster. In the case of the height raster generated with *k*-NN Random Forest, the presence of areas of pixels of a similar height (homogeneous patches) was detected in the interior of the forest. Moreover, in this raster, a large variation of heights (heterogeneous patches) was observed at the edge of the forest, which can be explained by the edge effect on growth. Trees at the forest edge tend to be more exposed to climatic conditions, such as wind damage. In addition, trees at the edge tend to use their energy for a greater production of lateral branches (greater crown volume), resulting in a lower height.

### 4.2. Site Index Estimation

Site quality determines the potential for the productivity of a forest, which refers to the timber volume yield for a stand at final rotation and is quantified by the SI [7]. ALS technology provides a highly accurate map of the canopy heights of forest areas, and a good SI estimation by means of models using the highest percentiles of the ALS point cloud is expected. This is because of the high correlation between DH that is included in the SI equation [7]. In this study, linear models for SI prediction had very low values of $R^2$ ($R^2$p99 = 0.09; $R^2$p95 = 0.08; $R^2$p90 = 0.07). Soil properties, climate variables, and management practices can all influence site index. In addition, there is a high correlation between the DH and SI. In addition, DH was highly related with the highest percentiles of the ALS point cloud, while SI is a projection of DH at harvest age. These results are concordant with those obtained in Brazil to estimate SI using non-linear mixed estimation models for *Eucalyptus grandis* [6]. In the study in question, Chapman–Richards equation was applied, which is very similar to the SI equation used in this work [7]. The *k*-NN Random Forest model had a $R^2$ = 0.65 and a RMSE = 1.62 m, similar to those obtained in Norway ($R^2$ = 0.69). Our model included the 99th percentile and the 80th bicentile, and the Norway model included the 90th, 60th bicentiles and the difference in the 90th height percentile ($\Delta$H90) [5]. In future studies, SI estimation may be improved by incorporating other variables such as soil properties, climate variables, and management practices [6].

### 4.3. Stand Segmentation

OTB segmentation has been proposed as a simple, more efficient, and more accurate method for stand delineation when compared to other more complex methods [17]. This automatic segmentation method is better than manual segmentation, principally owing to the faster generation of segments (stands). In this work, OTB segmentation method was applied based on a SI raster generated with a *k*-NN Random Forest model. The segmentation obtained had high homogeneity within each segment, in concordance with previous studies [6,17,39]. However, it is important to note that the shape and size of the resulting stands was not perfect, since there were forest areas which were not segmented, and areas with small polygons. These limitations have also been observed in previous studies and were manually modified [19]. Such imperfections are related to the combination of the parameters used and the raster of SI entered as input since the unsupervised Mean Shift (MS) classifier delineates the segments based on the digital levels of the raster. Different combinations of SR, RR and MRS were, therefore, assessed to overcome these restrictions. The optimal values obtained (SR = 30, RR = 1, MRS = 64, SR = 30) defined a minimum segment size of 2.03 ha, thus solving the generation of very small segments that were not representative of silvicultural conditions. The segmentation had 585 stands for the total surface (1995 ha) with an average area of 11.6 ha.

### 4.4. Forest Management Applications

The possibility of generating models to estimate SI based on ALS metrics is an important tool for intensive silviculture. When planning forest harvesting, it is essential to have the total volume at stand scale, with the DH being a fundamental variable for its calculation. ALS makes it possible to obtain this information from the whole study area using a low

number of calibration plots and generating more accurate results than using a traditional inventory. The availability of models to estimate height and SI using ALS metrics as an independent variable allows their application in large areas. OTB segmentation method improves stand delineation based on SI and is a very useful silvicultural tool to define forest productivity of different sites. Efficient forest management requires stands to be as homogeneous as possible within the forest in terms of SI. ALS segmentation obtained in this study had high intra-segment homogeneity and inter-segment heterogeneity. The average area of stands of 11.6 ha is consistent for the planning of both forests and harvests in intensive *Eucalyptus dunnii* plantations in Uruguay. The MRS parameter value defined a minimum stand of 2.03 ha, thus allowing relatively small stands to determine forest productivity in greater detail. Knowing forest productivity of a forest area will improve not only efficient harvesting, but also the planning of future plantations. Therefore, SI stand maps derived from ALS and OTB segmentation for *Eucalyptus dunnii* intensive plantations are an accurate option to improve forest planning.

## 5. Conclusions

This study shows that it is possible to generate models to estimate DH and SI using ALS metrics as independent variables for *Eucalyptus dunnii* intensive plantations. The best models for DH and SI estimation included the ALS metrics corresponding to the highest percentiles of the point cloud. It is worth noting that linear and *k*-NN Random Forest models had similar $R^2$ values to estimate DH. However, *k*-NN Random Forest had a slightly higher RMSE value. In terms of SI, it was not possible to fit linear models, but *k*-NN Random Forest model was fit. This is because there may be other variables that are significant in predicting site productivity (SI) that were not incorporated in the linear model. Soil properties, climatic variables and management practices are factors that can influence SI. These models allowed to obtain high spatial resolution maps of DH and SI for the entire study area. Automatic stand delineation obtained using SI improved the interpretation of potential productivity for each stand. This methodology provides an easy approach to update SI maps based on a raster derived from ALS metrics. Maps of SI are very useful for forest planning, as they can be used to define forest productivity at different scales, thus improving the decision-making process for forestry activities. This study, in addition to providing new tools for better forest management, promotes the need for further progress in the application of ALS data in intensive silviculture of *Eucalyptus* spp. plantations in Uruguay.

**Author Contributions:** Conceptualization, I.R.-M., R.N.-C. and A.H.-D.; methodology, I.R.-M., R.N.-C., M.Á.V.-M., A.H.-D.; formal analysis, I.R.-M., M.Á.V.-M., A.H.-D.; investigation, I.R.-M., R.N.-C., A.H.-D., R.A.-B.; resources, I.R.-M.; data cleansing, I.R.-M., A.H.-D., R.A.-B.; writing—original draft preparation, I.R.-M., R.N.-C.; writing—review and editing, all authors; supervision, R.N.-C.; project administration, I.R.-M.; funding acquisition, I.R.-M., R.N.-C. All authors have read and agreed to the published version of the manuscript.

**Funding:** This research was funded by SILVADAPT.NET (RED2018-102719-T), EVIDENCE (Ref: 2822/2021) and REMEDIO (PID2021-128463OB-I00).

**Data Availability Statement:** Data can be accessed by contacting the corresponding author.

**Acknowledgments:** We would like to acknowledge the support provided by SILVADAPT.NET (RED2018-102719-T).

**Conflicts of Interest:** The authors declare no conflict of interest.

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
