# Peer review of "Site Index Estimation Using Airborne Laser Scanner Data in Eucalyptus dunnii Maide Stands in Uruguay"

_forests, doi:10.3390/f14050933_

Round 1

Reviewer 1 Report (Previous Reviewer 2)

As this manuscript is submitted the second time, but without  marking the revised parts. So it is time-consuming to finish reviewing as I have to find where has been changed word by word myself.  Most concerns seem to have been addressed.

1P9L290 and P9L295: There are still no Table S1 and Table 5. Please correct.

2What is the role of Table 3? The gmsd values for p99, p75, and b80 are the same, so why is the highest percentile of the point cloud selected based on Table 3 What is the role of Table 3? The gmsd values for p99, p75, and b80 are the same, so why is the highest percentile of the point cloud selected based on Table 3. Please explain it carefully for readers to understand.

Author Response

The document "Reviewer_R1.docx" contains the modifications made to the manuscript.

Reviewer 2 Report (Previous Reviewer 1)

Dear authors

Congratulations. These modifications are satisfactory.

Author Response

Thank you for your time and dedication

Reviewer 3 Report (New Reviewer)

General comments:

This interesting paper assesses a method for predicting site index using ALS data in Eucalypt plantations in Uruguay. The topic is highly relevant because only limited research has been done on the use of ALS for site index estimation, and this area of research has great implications for forest inventory and management. The paper is well structured. The introduction provides a good overview of the background and motivation of this study, and the research gap it fills.

My main concern with this paper is the writing. In some parts of the manuscript, the writing is clear, concise, and correct. In other parts, the writing is unclear and it is difficult to understand exactly how the research was performed. I have provided some detailed comments on the writing and the grammatical errors in the abstract and the first few lines of the introduction, please see below. I could have continued providing comments on this issue, as there are grammatical errors throughout large parts of the manuscript. However, in my opinion, the writing in this manuscript is not fit, and it is difficult for a reviewer to evaluate the scientific work when the manuscript is not clearly and precisely written. I was trying to understand the methods from, for example, sections 2.4 and 2.6, and I can see that this is a very interesting study however I am not able to give a comprehensive review as the methods are not clearly explained. I therefore advise the authors to send the manuscript to an English language editor or a native speaker so that a more comprehensive review can be provided for a revised manuscript.

Specific comments

L17 The first two sentences of the abstract should grab potential readers’ attention, however they come a bit “out of the blue” as they do not represent a coherent justification for the current study. As it is now, I would remove these two sentences as they do not add any relevant information, or reformulate so that the sentences justify the study.

L19: Generate a stand scale “mapping” is not correct; mapping is a verb, perhaps the authors meant “generate stand scale maps”.

 L22: “Linear and non-parametric models with Random Forest imputation were fitted” The start of this sentence is incorrect, as it states that linear random forest imputation was fitted. But “imputation” can not be “fitted”, a model is fitted, and random forest models are not linear models. So this is confusing, and should instead be written as, for example: “Linear and Random Forest models were fitted …”

L24: “Linear models for dominant height (DH) estimation had a better fit (R2=0.84, RMSE=0.94 m, MAPE=0.039, Bias=0.002) than the Random Forest 25 (R2=0.85, RMSE=1.27 m, MAPE=7.20, Bias=-0.173)” here the reader will wonder why the model with the smallest R2 value is referred to as “a better fit”. If the RMSE was used as a statistic to assess the fit of the model, then the R2 is not relevant and should be removed.  However, on L179 it is stated that “(R2) was used to determine the goodness of fit” and so the statement in the abstract seems to be incorrect.

L27 at this point it has not been made clear to the reader what “RMSE/RMSE cross-validation” is.

 L29 With “99th percentile” I presume the authors meant “99th height percentile” (and not, for example, the 99th percentile of Longitudinal values, or laser intensity values) and I do not know what the 80th bicentile is.

L39 “around 4.77% is occupied by native forests 39 and 5.91% by planted forests”- the word “around” indicates a rough approximation, but the two decimal places of the percentage indicate a very precise estimate. Therefore, I suggest that “around” is removed, or the percentages are rounded to whole numbers.

L160: Are the authors sure that the function summarize() was used to obtain R2 and RMSE of the model? Normally, summarize() is applied to a vector, and summary() is applied to an lm object. However, summary() will normally not provide an RMSE when applied to a model object.

L163: Here, k-NN appears for the first time, without being introduced to the reader, and “with Random Forest distance calculation” is added. This is by no means a well-known modelling approach and should be explained clearly. For example, what does “random forest distance calculation” imply, are the distances based on proximity matrices obtained from the random forest model? If so, that should be explained. The study should be reproducible and the way this is explained, readers will not know how the “k-NN Random Forest” was applied.

L201: “the canopy height raster” appears for the first time, and the reader will not know what this is.

L203: “The pixel size of the ALS raster was determined by the square root of the area of the field plot.” Has this been done before in a previous study, which can perhaps be cited?

L204: " This raster had a value of the average canopy height (m) for each pixel.” What was the reason for the average canopy height being used for each pixel? I remember reading a paper:

Socha, J., Pierzchalski, M., Bałazy, R., & Ciesielski, M. (2017). Modelling top height growth and site index using repeated laser scanning data. Forest Ecology and Management406, 307-317.

In which the maximum laser height for each pixel may have been used (I may be mistaken), as it was found to best describe the top height.

 L207 Were ages for all trees in the data available, were the ages measured in the field? In that case, this should be explained under 2.2 – Field data. In that section, the registration of age of trees is not mentioned.

Author Response

The document "Reviewer_3_R2.docx" contains the modifications made to the manuscript.

Round 2

Reviewer 3 Report (New Reviewer)

General comments:

The scientific writing has improved since the last version, however further editing is needed. For example, the writing should be made more consistent (e.g., ALS/lidar/LiDAR, fit value/coefficient of determination, kNN models/kNN Random Forest). Terms are also not always introduced in the right way, many sentences are grammatically incorrect, and abbreviations are inconsistently used. In many cases, an abbreviation is introduced (e.g. SI, DH), but then the abbreviation is not used and the terms are still written out. I have considered and partially commented the writing, please see the specific comments, however I have focused this review mostly on the analyses and will leave the evaluation of the quality of the writing up to the journal.

Another general comment: dominant height is a main concept for the determination of SI, however it has not been defined in the manuscript. On L52 it is mentioned that the SI is based on the dominant height. Dominant height is typically defined as the mean height of the 100 tallest trees per ha, although other definitions have been proposed, such as the largest trees according to dbh (see, for example, Sharma et al. 2002). Which definition of dominant height is used here? Perhaps this could be explained, along with what is common in operational practices in Uruguay?

I would like to finally state that I enjoyed reading this manuscript. I had the impression that it was written by a talented early career scientist and some of the specific comments were formulated from this perspective.

I hope the comments will be helpful to improve the study.

Specific comments

L18 one of the best alternatives (alternatives should be plural)

L19 a stand-scale map (singular) or multiple stand-scale maps (plural).

L30 many readers will wonder what bicentiles are. This should be explained at least in the material and methods section (currently this term has not been introduced).

L64 although Spanish readers will understand the word “dasometric”, it does not appear in the oxford dictionary. I suggest reading this post and reformulating to something more common:

https://english.stackexchange.com/questions/513141/dasometry-is-this-a-common-word-in-english-is-there-more-common-alternative

L70 new paragraph starting with “Stand delineation”?

L82 site index –> SI (I suggest searching and replacing in the entire document, also for other abbreviations)

L90 I suggest “assess the SI on the basis of ALS dominant height and stand age” (the SI is not only based on height but also age).

L91 what are stands of forest productivity? (this sentence is incorrect).

L116 how was dominant height computed? Was it the mean height of the 3 tallest trees on the plot, or the largest trees regarding DBH? This should be explained here.

L130 Number of plots

Table 1. dbh- should this be the plot-wise mean values of dbh?

L133 Here, “ALS” is used, but in other places, “LiDAR” and “lidar” are used. Do these refer to the same thing? It would be good to be consistent.

L152 here the term “bicentile” should be introduced. “Percentile” can be considered a well-known term and needs no explanation. I am assuming the “bicentile” corresponds to the “bincentile” from Lastools, but this should be explained. In this case, I suggest to use the term “bincentile” and explain it in a way that corresponds to the Lastools documentation.

L154 what are estimation models? Would the term “predictive models” be correct? The term “estimation” should be reserved for when an estimator is used (please check this also in other parts of the manuscript).

L155 perhaps reconsider the word “conducted”, as models and correlation matrices cannot be conducted.

L156 here “predictive models” is used, are these the same models as the estimation models, and if not, what is the difference between the estimation models and the predictive models?

L158 I would suggest “models were fitted”.

L159 This still does not seem correct- perhaps the authors meant “the summary function”?

L175 “The k value… value had” this sentence should be edited as it is not grammatically correct.

L180 here, “the model” indicates that a single model was fitted, however in 2.4, two predictive models were described (the linear model and the kNN model).

L180 What is the “k-NN model”? Is it the same as “k-NN Random Forest” on L164? I suggest the authors to be consistent in the use of terms, to avoid many different versions of “estimation models”, “predictive models” “ALS predictive models”, “kNN models”, “kNN Random Forest” and “model”.

L181 “the range”?

L184 Here, R2 is introduced, but it also appears on L160 in abbreviated form. The abbreviation should be given the first time the term appears, and then used in abbreviated form for the rest of the manuscript.

L185 Predictive performance?

L195 space before k-NN

L212 Does the equation proposed by [7] use age at breast height as input (from core samples taken at breast height) or age since planting?

L213 I suggest “reference age” (consistent with L222) or “index age” instead of “age of the reference rotation”.

L213 Here it would be good to explain to the reader that the age was obtained from the stand register and based on the year of planting.

L222 I do not understand the “speciesX (years) part. Is the X perhaps a typing error?

L226 of a raster (singular) or of rasters (plural)

L228 SI

L244 “conducted” is not the right word here.

L246 “that with the highest variance was selected” is not grammatically correct.

L250 This sentence is grammatically incorrect, and “predict” should be used instead of “estimate”.

L254 R2 has already been introduced.

L261 “fit values” are mentioned, which can be replaced by “coefficients of determination” and here “(R2)” could be added to avoid repetition.

L261 what is mean by “and cross validation”, perhaps “results obtained in cross validation”?

L263 I think BIAScv should indicate systematic errors obtained in cross validation.

Figure 3 as well as other figures: it would be good to provide a higher resolution for improved quality.

L266 It would be good to make the figure caption consistent with the caption of Figure 3 (by including the model number).

Table 4 It is unusual to see a table with only one row. Would it be possible to combine Tables 2 and 4 to provide an overview of all the models, and then perhaps a separate table with the validation results?

L287 This paragraph states that Figure 3 “considers” a “significance of p-value<0.01” which is grammatically incorrect.

L288 This next sentence is also incorrect because values cannot determine anything.

L297 what were the criteria for determining that the SI “estimation” did not fit? (Also, “estimation” does not fit. A model fits). Was there no way of obtaining a valid model, for example by transformation?

L298 coefficient of determination -> R2

L327 – it would be interesting to see some results of the segmentation, for example the number of stands segmented, stand sizes, differences between numbers of stands obtained using the different segmentation methods.

L351 it would be good to specify which model had a “hight accuracy” (as there are multiple models) and stating that the accuracy was high would normally be relative to previous studies. There are many studies with accuracy assessments of DH models, it would be good to compare the results to those studies to support this statement.

 L347 lineal model?

L390 R2p99 has not been defined, this is the first occurrence in the manuscript, so the reader will not know what this is.

L391 Here, the abbreviation SI is re-defined as site productivity (previously in the manuscript it was site index) this is highly unusual.

L394 IS has not been defined.

L394 “In addition, … at harvest age” please always read the text carefully for checking the logic of every single statement.

L439 here the number of segmented stands appears for the first time, but this is actually a result which should be included in the results section, particularly when the result is then discussed.

L443 Also this sentence should be reformulated, considering the logic.

L446 It is best to refrain from any statements regarding the economics of the methods, when this has not been specifically assessed in the study.

L454 perhaps include why it was not possible to fit linear models.

L457 it is stated that the method is inexpensive, however this is the first time this is mentioned (in the conclusions) and it is not supported by any data or text in the above.

Author Response

The attached pdf contains the letter with the modifications made.

This manuscript is a resubmission of an earlier submission. The following is a list of the peer review reports and author responses from that submission.

Round 1

Reviewer 1 Report

The aim was to generate a stand-scale mapping of the site index for commercial plantations of Eucalyptus dunnii Maide using airborne laser scanner data. This research is valuable, but the current version is not yet ready for publication. Specific suggestions are as follows:

1.     It is difficult to see the outstanding strengths or contributions of your work compared to other work.

2.     The Meanshift algorithm is a well-known segmentation algorithm, but the authors may not really understand this algorithm or explain it clearly. Meanshift performs segmentation based on pixel locations, not Site Index. From a principle point of view, why Meanshift is effective for stand segmentation.

3.     Formula 1 is important and the highlight of your paper. How it is applied, however, is not clearly explained.

4.     In the abstract section, the context of your work should be added, so that readers can more easily understand the value of your work.

5.     The full name of ALS in the abstract and the text is different, which is confusing.

Author Response

Agradecemos los comentarios y sugerencias del editor y revisor, hemos revisado minuciosamente el manuscrito para incluir todas sus sugerencias. Encuentre a continuación nuestras respuestas a las preguntas específicas del revisor. Hemos revisado el manuscrito completo para mayor claridad, poniendo especial énfasis en resaltar la novedad y las implicaciones de nuestro trabajo.

Consulte el archivo adjunto

Reviewer 2 Report

This study generated a model that can estimate the height and site index, according to the ALS metrics as an independent variable, which has certain research significance for the evaluation of the production capacity of the site. The test and discussion are thorough while the novelty is relatively lacked. The presentation of the manuscript needs improvement. Detailed comments are as follows:

Some major points:

1.       The linear model has achieved good results in evaluating the height, while the introduction to the linear model in the methodology is not enough: how the linear model is generated; how the model parameters involved in Table 2 are obtained.

2.       The experimental results in Section 3.3 should be explained. The evaluation results of the linear model for SI are not satisfactory. Why is there a big difference between the prediction of dominant height (DH) and site index (SI)? Similarly, it’s necessary to give further explanation for the performance of Model 4 (k-NN Random Forest).

3.       Is there any difference between the selection of LiDAR metrics and the prediction of the height and site index? In Table 3, the good gmsd and sd are get when the candidate variable is b80. Why don’t you choose this variable to build a model for height prediction?

Some minor comments:

1.       The names of variables need to be unified. For example, "H" and "DH" are both expressions of height. However, "H" is used in the summary and conclusion, while "DH" is used in the experimental part.

2.       Abbreviations of professional terms should be indicated when they first appear in the paper, such as "spatial radius (SR)", and "minimum region size (MRS)".

3.       The results given in Lines 24 and 25 are not identical to the results in Tables 2 and 3. Please check carefully.

4.       What is the meaning of 'P' in Figure 2, please explain.

5.       P5L189: The pixel size is 17.8cm. How is it determined?

6.       P5L198: The format of Eq 1 needs to be modified.

7.       P6L229: Please explain what " the 99th, 95th and 90th percentiles" means.

8.       P6L238: What is the meaning of 'y', please correct.

9.       P7L248: 'q' should be italicized.

10.    P9L290 and P9L295: There are no Table S1 and Table 5. Please correct.

11.    "It is... similar results" is written in P12L418, but "When using... not fit satisfactorily" is written in P8L272. The two are contradictory, please modify them.

12.    Lines421-423. Hard to understand, please revise it.

Author Response

(The authors gave the same response as above.)
